# HPLC–DAD Analysis, SFE-CO_2_ Extraction, and Antibacterial Activity on Bioactive Compounds from *Mosla chinensis* Maxim

**DOI:** 10.3390/molecules28237724

**Published:** 2023-11-23

**Authors:** Ruixi Gao, Bingchen Han, Yanfeng Zeng, Linchuang Shen, Xinqiao Liu, Qiang Wang, Maochuan Liao, Jun Li

**Affiliations:** 1School of Pharmacy, South-Central Minzu University, Wuhan 430074, China; 2023021@mail.scuec.edu.cn (R.G.); wq@mail.scuec.edu.cn (Q.W.); 2College of Life Sciences, South-Central Minzu University, Wuhan 430074, China; 2023010071@mail.scuec.edu.cn; 3School of Life Sciences, Wuchang University of Technology, Wuhan 430223, China; 4Ethnopharmacology Level 3 Laboratory, National Administration of Traditional Chinese Medicine, South-Central Minzu University, Wuhan 430074, China

**Keywords:** *Mosla chinensis* Maxim, HPLC–DAD analysis, SFE-CO_2_, response surface methodology, antibacterial activity

## Abstract

*Mosla chinensis* Maxim is an annual herb with many potential purposes in agricultural, industrial, and pharmaceutical fields. At present, the extract of the whole plant from *M. chinensis* has been proven to demonstrate antifungal, antioxidant, and anti-inflammatory activities. Previous studies focused on the enzyme pretreatment in hydrodistillation from *M. chinensis*. However, organic solvent or supercritical fluid carbon dioxide extraction (SFE-CO_2_) methods, which are commonly utilized in industry, have seldom been studied and cannot provide multiple evaluations of yield. In this work, we analysed compounds from *M. chinensis* by HPLC–DAD, discussed *n*-hexane extraction, and conducted further investigations on SFE-CO_2_ through the design of response surface methodology (RSM). The sample obtained from pilot-scale SFE-CO_2_ was also tested against nine kinds of microorganisms. Single-factor results revealed that the extraction rates from *M. chinensis* by steam distillation, *n*-hexane extraction, and SFE-CO_2_ were 1%, 2.09%, and 3.26%, respectively. RSM results showed a significant improvement in extraction rate through optimising pressure and time, and the interaction of both factors was more important than that of temperature–pressure and temperature–time. A pilot-scale test with an extraction rate of 3.34% indicated that the predicted RSM condition was operable. In addition, samples from the pilot-scale SFE-CO_2_ showed antibacterial effects against three previously unreported bacteria (*Gardnerella vaginalis*, methicillin-resistant *Staphylococcus aureus*, and *Propionibacterium acnes*). These results fill the gap in previous research and provide more information for the application and development of *M. chinensis* in the future.

## 1. Introduction

*Mosla chinensis* Maxim is an annual herb belonging to the genus *Mosla* (Labiatae) and is widely distributed in the southern region of the Yangtze River of China [1,2]. The price of *M. chinensis* is stable and has been maintained at approximately USD 1 per kilogram for the past 9 years (2014–2022, Appendix A). The whole plant of *M. chinensis* has been proven to demonstrate antifungal, antioxidant, and antiviral activities that can be utilized in agriculture, industry, pharmacy, and other relevant fields [3,4,5]. Enzymatically assisted hydrodistillation of *M. chinensis* has been reported, and the potential development of the plant as an industrial crop has been highlighted [1].

However, to obtain bioactive compounds on a large scale and achieve industrial production, the extraction rate should be considered one of the most significant factors for evaluating the extraction method. Although some references report that enzyme pretreatment could increase compound production [1,6], the extraction rate of hydrodistillation may still be lower than that of organic solvent or supercritical fluid carbon dioxide extraction (SFE-CO_2_) methods, leading to inefficiency in large-scale or industrial production. In addition, few experiments on organic solvent or SFE-CO_2_ from *M. chinensis* have been reported; therefore, it is necessary to conduct relevant studies to compare the extraction rates among the three methods. In addition, gas chromatography (GC) is commonly used for detection compounds due to its high sensitivity and selectivity [7,8]. However, under certain conditions (limited funds, workplaces, and staff), it may not be possible to immediately use GC to detect components in factories or pilot workshops. Therefore, it would be beneficial to find suitable liquid chromatography (LC) methods, such as thin-layer chromatography (TLC) and high-performance liquid chromatography (HPLC), to detect the components.

In this report, we discussed steam distillation and *n*-hexane extraction of compounds from *M. chinensis* and further investigated SFE-CO_2_ through the design of response surface methodology (RSM) to compare the extraction rates of the three methods. The optimised SFE-CO_2_ extraction condition was verified by a pilot-scale experiment, and the sample obtained from pilot-scale SFE-CO_2_ was also tested against nine kinds of microorganisms (including bacteria and fungi). In addition, TLC and HPLC were used in this study to provide alternative analytical methods.

## 2. Results and Discussion

### 2.1. Chemical Analysis of M. chinensis by TLC and HPLC–DAD

The advantages of TLC include not relying on instruments, simple operating conditions, and fast results. To date, TLC has remained one of the most important separation and analytical methods for natural products [9,10]. Through TCL analysis, the main components thymol and carvacrol could be analysed within 15 min under the developing solvent of *n*-hexane:toluene:acetone:acetic acid (2:8:0.5:0.5, *v*/*v*) and filmed by a 5% vanillin-sulfuric acid developer (Appendix A), providing a potentially fast method for analysis in factories or pilot plants.

Like GC, HPLC also has applications in analysis of natural products [11,12]. In this study, the whole plants of *M. chinensis* were collected from 16 places to select the species with the highest chemical composition content before the next extraction process. The detector wavelength was set at 274 nm according to DAD results when screening plants, because the main compounds carvacrol and thymol had the maximum UV absorption (Figure 1A). Under gradient elution conditions (methanol (A):water (B) = 60:40–65:35, 25 min), the compounds carvacrol and thymol in the extract from *M. chinensis* could be detected within 25 min, providing an effective analytical method. Considering the sum areas of carvacrol and thymol, sample 3 (from Bozhou, Anhui province, China, Figure 1B,C) was used for the next extraction experiment.

Next, samples obtained from the three extraction methods (steam distillation, *n*-hexane extraction, and SFE-CO_2_) of *M. chinensis* were compared by HPLC. The wavelength was set at 210 nm to ensure the standard compounds used in this study could be detected (only at 210 nm could terpinen-4-ol, *p*-cymene, *γ*-terpinene, and humulene have good UV absorption) when comparing the components obtained from the three extraction methods. Benefiting from standard compounds as references (Figure 2A) and results recorded by a DAD, the components (terpinen-4-ol, carvacrol, thymol, *p*-cymene, *γ*-terpinene, and humulene) of *M. chinensis* were separated by gradient elution with acetonitrile and water (Figure 2B–E), and the proportions of the main components were shown in the form of relative peak areas (Table 1). Although the content ratios of some compounds (terpinen-4-ol, *p*-cymene, *γ*-terpinene, and humulene) changed due to the low sensitivity of the HPLC detector compared to GC, as well as impurities from *n*-hexane extraction and SFE-CO_2_, carvacrol (retention time: 13.87 min) and thymol (retention time: 14.773 min) were still the main components (Figure 2B), contributing more than 80% of the total content (210 nm, Table 1), which was similar to the GC results reported in the literature [1]. In general, taking HPLC as an analytical method for *M. chinensis* was feasible.

### 2.2. Solid–Liquid Ratio and Extraction Time Results for Steam Distillation and n-Hexane Extraction Rate

As shown in Figure 3A, the extraction rate by steam distillation and *n*-hexane extraction was positively correlated with the extraction time. Within a given time range (2 h to 6 h), the extraction rate of steam distillation increased from 0.94% to 1.00%, and the extraction rate of *n*-hexane increased from 1.81% to 2.08%. The solid–liquid ratio of both extraction methods was not significantly related to the extraction rate, as shown in Figure 3B. As the solid–liquid ratio increased, there was no change in the extraction rate for steam distillation, and there was only a 0.05% rate improvement for *n*-hexane extraction. Thus, the optimization of extraction time contributed to the yield, while the difference in extraction rates between steam distillation and *n*-hexane extraction mainly depended on the extraction method itself.

Other factors, such as extraction temperature, could theoretically be set as a test factor, as the higher the extraction temperature is, the faster the rate of steam generation. However, it was difficult to carry out such experiments. On the one hand, the steam distillation method has an extraction temperature of approximately 95 °C, and the boiling point of pure water is 100 °C, leaving a harsh inspection range of 5 °C. On the other hand, an excessive extraction temperature could generate a large amount of steam, which could diffuse into the air or overflow into the heating device, leading to accidents, especially in conditions with low efficiency of cooling water. This phenomenon was particularly prone to occur during organic solvent extraction. Therefore, only two factors were examined in the single-factor experiments and the selected extraction parameters were set as follows: solid–liquid ratio = 1:8, extraction time = 6 h (for steam distillation); solid–liquid ratio = 1:12, extraction time of 5 h (for *n*-hexane extraction), and the corresponding extraction rates were 1.00% and 2.09%, respectively.

### 2.3. Optimization Results of SFE-CO_2_ Extraction by Single Factor and RSM

#### 2.3.1. Single-Factor Results of Temperature, Pressure, and Time for Extraction Rate

Because of the influence on the physical properties of CO_2_ (supercritical pressure = 7.39 MPa at a temperature of 31.26 °C), changing pressure and temperature may affect the diffusion rate, density, and viscosity of CO_2_ molecules as well as the extraction rate. As shown in Figure 3C, the increased temperature (35 °C, 40 °C, and 45 °C) could accelerate the molecular motion rate, enhance the mass transfer ability, and improve the extraction rate of EO from 2.63% to 3.26%. However, the excessive temperature could also decrease the density of supercritical CO_2_ and the solvation ability. From 45 °C to 55 °C, the CO_2_ density reduced more quickly, leading to a decline in the extraction rate from 3.26% to 2.61%. Thus, the extraction temperature was selected to be 45 °C.

The impact of pressure changes was more significant, as shown in Figure 3D. When the extraction pressure increased from 10 MPa to 15 MPa, the elevated CO_2_ density was beneficial in improving the mass transfer ability and extraction efficiency. Therefore, the extraction rate doubled from 1.25% to 2.69%. Similar to temperature, the excessive pressure could cause a decreased viscosity of CO_2_ and reduce the mass transfer rate, resulting in low extraction rates. In addition, a higher pressure could synchronously lead to a larger instrument bearing capacity and operational risk. Therefore, the extraction pressure was determined to be 15 MPa.

For the extraction time, there was a rapid positive correlation with the extraction rate within the range from 0.5 h to 1.5 h, increasing from 0.7% to 2.66%, as shown in Figure 3E. After 1.5 h, the longer extraction time contributed slowly to increasing the extraction rate. Considering the sealing of instruments and the gradual flow loss of CO_2_ during the extraction process, 1.5 h was used for the next stage of RSM optimization.

#### 2.3.2. The RSM Results, Variance Analysis, and Verification of SFE-CO_2_

RSM is an effective means of solving practical questions [13], not only for bringing in random errors and fitting complex unknown functional relationships in a small area through a calculated first- or second-order polynomial model, but also for continuously analysing various levels of experiments during the optimization process compared to orthogonal experiments. Thus, RSM was suitable for evaluating the extraction rate of crops [14]. In this study, based on the results of the single-factor experiments, the investigated ranges for temperature, pressure, and time were 40–50 °C, 10–20 MPa, and 1–2 h, and the corresponding intermediate values were 45 °C, 15 MPa, and 1.5 h according to the design principles of the Box–Behnken central combination experiment. RSM optimization was designed using Design Expert software (version 8.0.6), with the extraction rate (Y) as the response value and temperature (A), pressure (B), and time (C) as the response factors. The model regression equation was Y = −50.68575 + 1.7376A + 1.37245B + 4.8595C + 0.0004AB − 0.008AC + 0.041BC − 0.01929A^2^ − 0.04619B^2^ − 1.479C^2^.

The variance analysis results are recorded in Table 2. The significance of the coefficients of each variable in the regression equation were checked by *F*-tests. From the variance analysis, it is shown that the *p*-value of the regression model is extremely significant (*p* < 0.0001). The *p*-value of the lack-of-fit item is 0.2816 (*p* > 0.05), indicating that unknown factors have little influence on the experimental results and introducing higher-order terms is unnecessary for this model. Therefore, the regression model can fit well with the experimental data and reflect the effects of temperature (A), pressure (B), and time (C) on the extraction rate. According to the *F*-values in Table 2, factors like pressure (B) and time (C) and BC, A^2^, B^2^, and C^2^ have an extremely significant impact on the extraction rate (*p* < 0.001). The *F*-value of BC (21.08) is much higher than those of AB and AC, indicating that the interaction effect of extraction pressure and time is the most important factor changing the extraction rate. Therefore, the order of factors affecting the extraction rate is listed as follows: pressure ≈ time > temperature.

To further investigate the interaction effects of temperature, pressure, and time on the extraction rate, a 3D response surface diagram and contour figures deriving from the regression equation were plotted, as shown in Figure 4. The steep shape of the response surface and elliptical contour lines with a larger curvature reflect the influence of factor variables on the response value and the significance of the interaction between factors. Figure 4C shows a denser contour line along the pressure axis than along the temperature axis (Figure 4A) and time axis (Figure 4E), indicating that the extraction pressure had a greater impact on the response value than temperature and time. The elliptical contour lines in Figure 4D exhibit a more significant interaction between pressure–time than between temperature–pressure (Figure 4B) and temperature–time (Figure 4F) on the oil extraction rate, which is consistent with the results in Table 2.

The predicted ideal extraction rate (3.36%) based on RSM results by Design Expert software (version 8.0.6) was verified through a pilot-scale test. According to the optimal conditions (temperature = 44.84 °C, pressure = 15.82 MPa with 1.72 h extraction time), the actual experiment was carried out under the following conditions: temperature = 45 °C, pressure = 15.8 MPa with a 1.7 h extraction time. The pilot-scale extraction rate was 3.34%, i.e., after extracting 10 t of plants with the extraction parameters mentioned above, 334 kg of products was obtained (Appendix A). The fitting rate is 99.4% when compared with the predicted value, suggesting that the optimized scheme is reasonable and effective.

### 2.4. Antibacterial Activity of Extract Sample from SFE-CO_2_

As reported by references, antimicrobial activity is the main function discovered in bioactive compounds from *M. chinensis* [15,16,17]. Therefore, we continued to explore antibacterial tests on strains that have not been studied in order to expand the application of *M. chinensis* in other fields. The sample from the pilot-scale extraction was screened against nine microorganisms (seven bacteria and two fungi, Table 3) through the inhibition circle method at a concentration of 32 μg/mL. The results suggested that the sample from the pilot-scale SFE-CO_2_ extraction experiment could inhibit five bacteria (*Gardnerella vaginalis*, *Propionibacterium acnes*, *Staphylococcus aureus*, *Staphylococcus epidermidis*, and methicillin-resistant *Staphylococcus aureus*, Table 3, Appendix A), and the antibacterial circle diameters (mm) were 13.71 ± 0.49, 14.17 ± 0.17, 15.11 ± 0.78, 12.55 ± 0.53, and 14.89 ± 0.78, respectively. No inhibition circles were found for the four remaining microorganisms (*Helicobacter pylori*, *Malassezia furfur*, *Aspergillus flavus*, and *Epidermophyton floccosum*), possibly due to insufficient concentrations according to a literature report [3]. The reason for setting the concentration to 32 μg/mL was that if the concentration was too high, it would not distinguish which bacteria the sample had a significant effect on; if the concentration was too low, it might not exhibit any activity. Therefore, in this report, the screening concentration was set as 32 μg/mL.

After the antimicrobial circle screening, the MICs of sample against five bacteria were determined. In an experiment containing a positive group (tetracycline), the MIC values against the five bacteria were 16, 32, 16, 32, and 16 μg/mL (Table 3). Among these five bacteria, MRSA is a drug-resistant bacterium, *P. acnes* is associated with acne, and *G. vaginalis* is related to vaginitis. Among the compounds in *M. chinensis*, the bioactive thymol and carvacrol had significant antimicrobial activity which inhibited microbial growth by destroying the cellular structure (cell membrane) of microorganisms, resulting in the leakage of essential molecules such as protein and K^+^ [18]. Carvacrol has demonstrated relevant clinical antibacterial activity, for both Gram-positive and Gram-negative bacteria, and a synergistic effect when associated with gentamicin [19]. Therefore, the antibacterial results provide valuable information for the development of relevant drugs and cosmetics originating from *M. chinensis* in the future.

## 3. Materials and Methods

### 3.1. Chemical Reagents, Apparatus, and Materials

An ultrapure water system was purchased from Chunjie Science and Technology (Chengdu, China). Toluene, *n*-hexane, acetone, acetic acid, dimethyl sulfoxide (DMSO), HPLC-grade methanol, and acetonitrile were purchased from Sinopharm Chemical Reagent Co. Ltd., Shanghai, China.

TLC was performed on silica gel GF254 (Qingdao Haiyang Chemical Group Co., Qingdao, China). HPLC was performed on an Agilent 1200 series system (Agilent, Tokyo, Japan) equipped with a quaternary solvent delivery system, an autosampler, a column compartment, and a diode array detector (DAD). An MS-II-C18 column (250 mm × 4.6 mm, 5 µm, Cosmosil Co., Ltd., Kyoto, Japan) was used for HPLC analysis. SFE-CO_2_ extraction was performed on an RZSCF231-50 machine system (Nantong Wisdom Supercritical Science & Technology Development Co., Ltd., Nantong, China).

The whole plants of *M. chinensis* were collected from 16 places to select the species with the highest chemical composition content. The 16 regions were (1–16): Anguo (Hebei, 1); Nanjing (Jiangsu, 2); Bozhou (Anhui, 3); Huoshan (Anhui, 4); Chengdu (Sichuan, 5); Shennongjia (Hubei, 6); Panan (Zhejiang, 7); Quzhou-1 (Zhejiang, 8); Quzhou-2 (Zhejiang, 9); Yichun (Jiangxi, 10); Sanming (Fujian, 11); Kunming (Yunnan, 12); Guangzhou (Guangdong, 13); Jieyang (Guangdong, 14); Meizhou (Guangdong, 15); and Yulin (Guangxi, 16). Considering the sum areas of carvacrol and thymol, the sample from Bozhou (Anhui province, China) was used for the next extraction experiment.

### 3.2. Solid–Liquid Ratio and Time Investigation for Steam Distillation and n-Hexane Extraction

The investigation ranges for both methods were the same (the time was 2–6 h, and the solid–liquid range was 1:8–1:12). To choose the solid–liquid ratio, the extraction time was fixed (6 h for both methods) and the solid–liquid ratio was fixed (1:10 for steam distillation and 1:12 for *n*-hexane extraction) when selecting the extraction time. The yield was calculated by dividing the quality of the extraction compounds and the masses of plants (50 g).

### 3.3. SFE-CO_2_ Extraction Results for Bioactive Compounds from M. chinensis

#### 3.3.1. Single-Factor Results of Extraction Temperature, Pressure, and Time

The design idea was the same as that of steam distillation and *n*-hexane extraction. Only one factor was investigated with the other two factors unchanged. The default extraction parameters (temperature, pressure, and time) were fixed at 40 °C, 15 MPa, and 1.5 h, respectively, and the CO_2_ flow rate was set at 35 kg/h.

#### 3.3.2. Response Surface Optimization

Based on the principle of the Box–Behnken central combination experiment and the results of the single-factor test, RSM was designed with the extraction rate as the evaluation target and temperature, pressure, and time as independent variables. Seventeen experimental factors and levels are shown in Table 4. Analysis of variance was used to determine the significant differences in the extraction rate under various conditions.

### 3.4. Chemical Composition Analysis

The powder of *M. chinensis* (1 g) was extracted with *n*-hexane, dissolved in DMSO, and treated with 0.45 μm membrane filtration for HPLC analysis. When selecting the plant species with the highest chemical composition content, the HPLC gradient elution ratio was methanol (A): water (B) = 60:40–65:35, 25 min. Test samples were dissolved in DMSO at equal volume and the injection volume was 3 μL. Chromatographic peaks and areas of carvacrol and thymol from 16 places were compared at 274 nm because the most important compounds carvacrol and thymol had the maximum UV absorption.

When analysing samples from the three extraction methods, the HPLC elution condition was acetonitrile (A): water (B) = 45:55–100:0, 55 min, and each sample was dissolved in methanol, fixed in a volumetric flask (10 mL), and diluted with methanol at equal volume. Standard compounds used for HPLC comparison were purchased from Aladdin Company, Shanghai, China. All injection volumes were 3 μL. The sample powder (50 g) was filtered through a 20-mesh sieve and the wavelength was set at 210 nm to ensure the standard compounds used in this study could be detected (only at 210 nm could terpinen-4-ol, *p*-cymene, *γ*-terpinene, and humulene have good absorption) when comparing the components obtained from three extraction methods.

### 3.5. Antibacterial Activity of Bioactive Compounds from M. chinensis

#### 3.5.1. Microbial Strains

The sample from SFE-CO_2_ extraction was tested on a panel of microorganisms, which included the following laboratory control strains obtained from the American Type Culture Collection (ATCC, Rockville, MD, USA): *Propionibacterium acnes* (ATCC 11827), *Staphylococcus aureus* (ATCC 25923), *Staphylococcus epidermidis* (ATCC 35984), methicillin-resistant *Staphylococcus aureus* (MRSA, ATCC 43300), *Helicobacter pylori* (ATCC 43504), *Malassezia furfur* (ATCC 14521), *Aspergillus flavus* (ATCC 10836), and *Epidermophyton floccosum* (ATCC 52066) and the following strain from the BeNa Culture Collection (Beijing, China): *Gardnerella vaginalis* (BNCC 354890).

#### 3.5.2. Antimicrobial Circle Screening

The Kirby–Bauer agar disc diffusion method was used to evaluate the antimicrobial activity of the samples [20]. In brief, test antimicrobials were seeded on a culture plate (10^6^ CFU/mL) and incubated overnight at 37 °C until obvious bacterial plaques were visible. Then, the visible colonies were incubated in 5 mL of Luria–Bertani liquid medium at 37 °C for 8 h (for the anaerobic bacteria, incubation time was 48 h with Brain Heart Infusion liquid medium; for the fungi, incubation time was 72 h at 27 °C), reaching an OD 600 value of approximately 0.5; they were centrifuged and resuspended in PBS to a McFarland scale of 0.5. A sterile cotton swab was dipped in the adjusted antimicrobial solution and the solution was evenly coated on a plate. Then, the disks were impregnated with 32 μg/mL of extract, and the antimicrobial activity was evaluated by measuring the circle of inhibition against the test organisms.

#### 3.5.3. Determination of the Minimum Inhibitory Concentration (MIC)

The MICs of the samples on the test bacterial strains were determined by the microwell dilution method [21]. Inocula of the microorganisms were prepared from 8 h cultures (for the anaerobic bacteria, the incubation time was 48 h) and the suspensions were adjusted to 10^6^ CFU/mL bacteria. The EO was dissolved in 10% DMSO, and serial twofold dilutions of the extract samples were prepared in a 96-well plate, ranging from 1 μg/mL to 128 μg/mL. The MICs of the positive standard (tetracycline) were also determined in parallel experiments to control for the sensitivity of the microorganisms. The microorganism growth was indicated by turbidity and the MIC was defined as the lowest concentration of the sample at which the microorganism did not demonstrate visible growth.

### 3.6. Statistical Analysis

RSM design and variance analysis were carried out by Design Expert software (version 8.0.6), and antibacterial circle diameter was expressed as the mean ± standard deviation. The graphs were plotted with Origin software (version 2015).

## 4. Conclusions

In this study, HPLC was successfully used to detect the main components (carvacrol, thymol, and other volatile compounds) from *M. chinensis* and analysed the samples extracted from steam distillation, *n*-hexane, and SFE-CO_2_. Single-factor results for steam distillation and *n*-hexane extraction suggested that there was a certain positive correlation between the extraction time and yield. The two methods could achieve extraction rates of 1% (for steam distillation) and 2.09% (for *n*-hexane extraction). Furthermore, the extraction parameters (temperature = 45 °C, pressure = 15 Mpa, and time = 1.5 h) for SFE-CO_2_ extraction were also selected by single-factor optimization, and variance analysis together with RSM results indicated that the pressure and time contributed to the extraction rate, and the interaction of both factors was more important than that of temperature–pressure and temperature–time. After RSM optimization, SFE-CO_2_ extraction was verified through a pilot-scale experiment with a yield of 3.34%. The sample from the pilot-scale extraction showed antibacterial effects against *G. vaginalis*, MRSA, and *P. acnes*. All of these results make it clear that SFE-CO_2_ extraction is practical for achieving a relative higher yield of extraction samples with antibacterial activities, providing a basis for existing research as well as future development of *M. chinensis* and other crops.

## Figures and Tables

**Figure 1 molecules-28-07724-f001:**
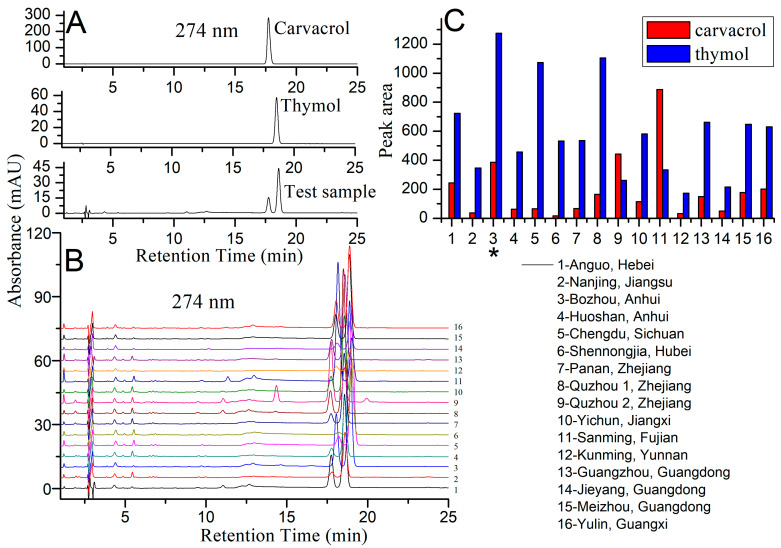
Comparison of carvacrol and thymol from 16 plant by HPLC. (**A**). HPLC results of carvacrol, thymol and test sample, and the detector wavelength was 274 nm. (**B**). HPLC results of *M. chinensis* from 16 places, and the detector wavelength was 274 nm. (**C**). Content comparison of carvacrol and thymol from 16 places. *, the species with the highest total content of carvacrol and thymol.

**Figure 2 molecules-28-07724-f002:**
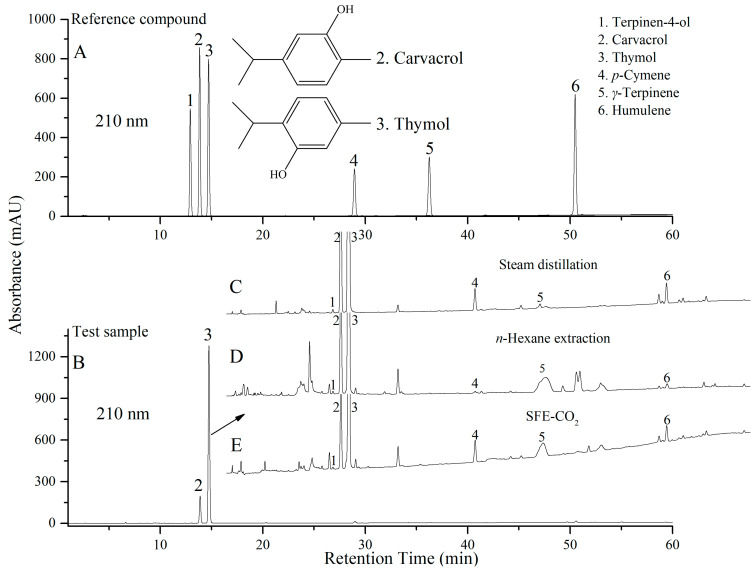
HPLC analysis of reference compounds (**A**) and *Mosla chinensis* (**B**) by steam distillation (**C**), *n*-hexane extraction (**D**), and SFE-CO_2_ (**E**).

**Figure 3 molecules-28-07724-f003:**
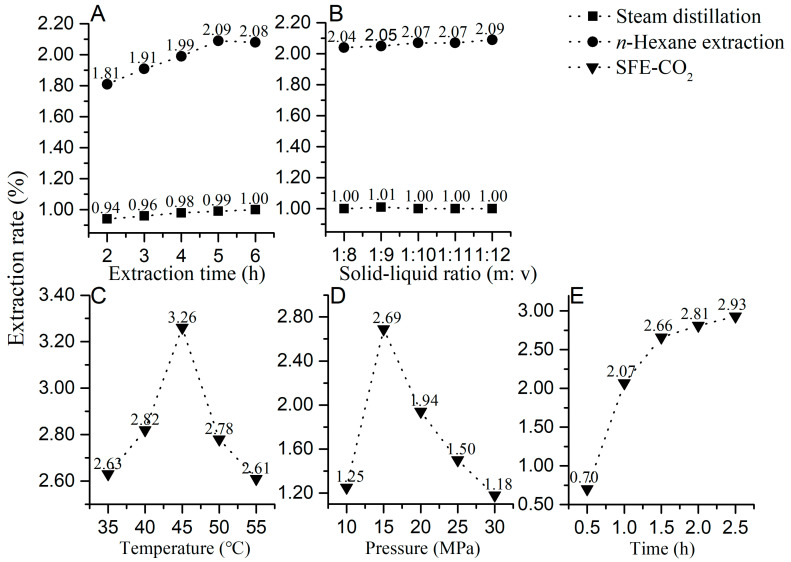
Single-factor results of three extraction methods for *Mosla chinensis*. (**A**,**B**) extraction time and solid–liquid ratio results for steam distillation and *n*-hexane extraction rate. (**C**–**E**) extraction temperature, pressure and time results for SFE-CO_2_ extraction rate.

**Figure 4 molecules-28-07724-f004:**
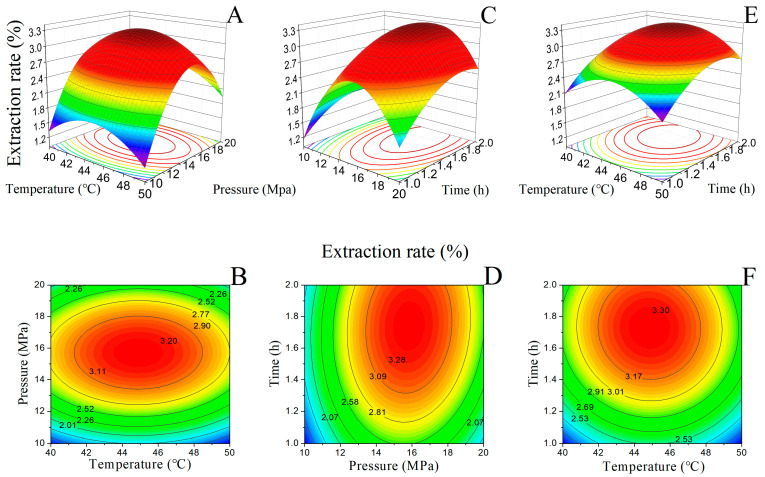
Response surface methodology results of SFE-CO_2_ for *Mosla chinensis*. For (**A**,**C**,**E**), the X- and Y- axes represent the extraction factors, and the Z- axis represents the extraction rate. For (**B**,**D**,**F**) the contour lines represent extraction rates.

**Table 1 molecules-28-07724-t001:** Main chemical compositions of *Mosla chinensis* ^a^.

No.	Component	Molecular Formula	CAS Number	Retention Time (min)	Relative Content Ratio ^b^ (%)		
					Steam Distillation	*n*-Hexane Extraction	SFE-CO_2_
**1**	Terpinen-4-ol	C_10_H_18_O	562-74-3	12.973	0.13	0.12	0.12
**2**	Carvacrol	C_10_H_14_O	499-75-2	13.87	12.03	10.62	8.13
**3**	Thymol	C_10_H_14_O	89-83-8	14.773	82.21	69.58	74.96
**4**	*p*-Cymene	C_10_H_14_	99-87-6	28.994	1.03	0.2	1.83
**5**	*γ*-Terpinene	C_10_H_16_	99-85-4	36.313	0.34	4.93 ^c^	4.27 ^c^
**6**	Humulene	C_15_H_24_	6753-98-6	50.562	0.92	0.33	1.29
Total identified components					96.67	85.78	90.6

^a^ Test samples were obtained from the following extraction conditions: 1. steam distillation: solid–liquid ratio = 1:8, extraction time = 6 h; 2. *n*-hexane extraction: solid–liquid ratio = 1:12, extraction time = 5 h; 3. SFE-CO_2_: temperature = 45 °C, pressure = 15 MPa, time = 1.5 h. ^b^ The detector wavelength was 210 nm. ^c^ Mixed with other components.

**Table 2 molecules-28-07724-t002:** ANOVA results of SFE-CO_2_.

Source	Sum of Squares	Degree of Freedom	Mean Square	*F*-Value	*p*-Value	
Model	9.6	9	1.07	535.18	<0.0001	significant
A—temperature	0.00405	1	0.00405	2.03	0.1971	
B—pressure	0.88	1	0.88	440.32	<0.0001	
C—time	0.92	1	0.92	460.49	<0.0001	
AB	0.0004	1	0.0004	0.2	0.6677	
AC	0.0016	1	0.0016	0.8	0.4001	
BC	0.042	1	0.042	21.08	0.0025	
A^2^	0.98	1	0.98	491.19	<0.0001	
B^2^	5.61	1	5.61	2816.31	<0.0001	
C^2^	0.58	1	0.58	288.75	<0.0001	
Residual	0.014	7	0.00199			
Lack of fit	0.00808	3	0.00269	1.83	0.2816	not significant
Pure error	0.00588	4	0.00147			
Cor total	9.62	16				

**Table 3 molecules-28-07724-t003:** Antibacterial activity of extracts from *Mosla chinensis* ^a^.

Name	Antibacterial Circle Diameter (mm) ^b^	MIC ^c^ (µg/mL)	
			Tetracycline
*Gardnerella vaginalis*	13.71 ± 0.49	16	2
*Propionibacterium acnes*	14.17 ± 0.17	32	1
*Staphylococcus aureus*	15.11 ± 0.78	16	<1
*Staphylococcus epidermidis*	12.55 ± 0.53	32	<1
methicillin-resistant *Staphylococcus aureus*	14.89 ± 0.78	16	<1
*Aspergillus flavus*	- ^d^		
*Epidermophyton floccosum*	-		
*Helicobacter pylori*	-		
*Malassezia furfur*	-		

^a^ The sample was obtained from a pilot-scale SFE-CO_2_ extraction experiment. ^b^ The experiment was repeated nine times. ^c^ MIC: Minimal inhibitory concentration. ^d^ -: No antibacterial circle was found at this concentration.

**Table 4 molecules-28-07724-t004:** Response surface methodology design for SFE-CO_2_ extraction.

Run	Factor			
	Temperature (°C)	Pressure (MPa)	Time (h)	Extraction Rate (%)
1	50	10	1.5	1.23
2	45	15	1.5	3.2
3	40	20	1.5	1.98
4	50	20	1.5	1.97
5	45	20	1.0	1.6
6	45	15	1.5	3.23
7	50	15	1.0	2.04
8	40	15	1.0	2.06
9	40	15	2.0	2.8
10	40	10	1.5	1.28
11	45	15	1.5	3.3
12	45	20	2.0	2.46
13	45	15	1.5	3.26
14	45	15	1.5	3.27
15	45	10	2.0	1.65
16	45	10	1.0	1.2
17	50	15	2.0	2.7

## Data Availability

Data are contained within the article and Appendix A.

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
