# Peer review of "HPLC–DAD Analysis, SFE-CO2 Extraction, and Antibacterial Activity on Bioactive Compounds from Mosla chinensis Maxim"

_molecules, 2023, doi:10.3390/molecules28237724_

Round 1

Reviewer 1 Report

Comments and Suggestions for Authors

This manuscript by Gao et al. discussed about HPLC-DAD analysis, SFE-CO2 extraction and antibacterial activity on bioactive compounds from mosla chinensis maxim. Authors analyzed different compounds from M. chinensis by HPLC-DAD, discussed n-hexane extraction and conducted further investigations on SFE-CO2 through the design of response surface methodology. The SFE-CO2 extraction conditions were verified by a pilot-scale experiment and tested for antimicrobial and antifungal activity. This manuscript may be suitable for publication in molecules journal after below correction.

1.     Abstract can be reduced to one paragraph.

2.   Did authors run reference standards (shown in CAS numbers; table 1) to match with constituents obtained from M. chinensis either by TLC or HPLC-DAD?

Author Response

Reviewer 1:

1). Abstract can be reduced to one paragraph.

Reply: Thank you for your suggestion. We have made correction according to your suggestion (Abstract, line 22).

2): Did authors run reference standards (shown in CAS numbers; table 1) to match with constituents obtained from M. chinensis either by TLC or HPLC-DAD?

Reply: Thank you for your comments. We conducted the experiments referred by your comments. In our manuscript, the results in Fig 2 were consistent with those in Table 1, namely, Fig. 2A showed the chromatographic results of each reference compound under HPLC analysis conditions, Fig. 2B showed the results of three extraction methods (steam distillation, n-hexane extraction and SFE-CO2 extraction); Fig. 2C, 2D, and 2E were enlarged results of the three extraction methods. The retention time of each reference compound in Fig. 2A was also corresponded to the data in Table 1. We have added image captions to the revised manuscript according to your comments (Line 91-92).

In addition, TLC was performed and uploaded in the supplementery data (S1), and shown in the attachment.

Reviewer 2 Report

Comments and Suggestions for Authors

Manuscript "HPLC-DAD analysis, SFE-CO2 extraction and antibacterial activity on bioactive compounds from Mosla chinensis maxim" shows that different extraction methods can influence the proportion of compounds with biological interest. However, the authors must add:

- The HPLC spectra of the three extracts obtained by different methods.

- Why antibacterial activity tests were not carried out on the other two extracts. This would be interesting to determine if extraction methods can interfere with the composition and proportion of Mosla chinensis maxim compounds.

- Why the bacterial species Propionibacterium acnes, Staphylococcus aureus, Staphylococcus epidermidis, methicillin-resistant Staphylococcus aureus, Helicobacter pylori, Aspergillus flavus, Epidermophyton floccosum and Gardnerella vaginalis were chosen.

- The authors in the discussions section could suggest the possible mechanism of action of the extract, commenting on the synergy of the compounds.

Author Response

Reviewer 2:

Manuscript "HPLC-DAD analysis, SFE-CO2 extraction and antibacterial activity on bioactive compounds from Mosla chinensis maxim" shows that different extraction methods can influence the proportion of compounds with biological interest. However, the authors must add:

1). The HPLC spectra of the three extracts obtained by different methods.

Reply: Thank you for your suggestions. We conducted the experiments referred by your comments. In our manuscript, the results in Fig 2 were consistent with those in Table 1, namely, Fig. 2A showed the chromatographic results of each reference compound under HPLC analysis conditions, Fig. 2B showed the results of three extraction methods (steam distillation, n-hexane extraction and SFE-CO2 extraction); Fig. 2C, 2D, and 2E were enlarged results of the three extraction methods. The retention time of each reference compound in Fig. 2A was also the same with the data in Table 1. We have added image captions to the revised manuscript according to your comments (Line 91-92).

2). Why antibacterial activity tests were not carried out on the other two extracts. This would be interesting to determine if extraction methods can interfere with the composition and proportion of Mosla chinensis maxim compounds.

Reply: We did not conduct antibacterial tests by samples obtained from other two extraction methods for the following reasons:

  1. There was no significant chemical component change amongthe samples obtained by the three extraction methods, so the antibacterial results of the three samples might be the same.
  2. According to a reference (Li et al., Industrial Crops & Products, 2022, 189: 115871), the antibacterial resultsof samples from different processing methods were nearly the same. 

Therefore, duplicating antibacterial tests on relevant samples may be unnecessary.

3). Why the bacterial species Propionibacterium acnes, Staphylococcus aureus, Staphylococcus epidermidis, methicillin-resistant Staphylococcus aureus, Helicobacter pylori, Aspergillus flavus, Epidermophyton floccosumand Gardnerella vaginalis were chosen.

Reply: The antibacterial effect of the extract from M. chinensis has been confirmed (Cao et al., Food Chemistry, 2009, 115: 801-805), therefore we continued to explore antibacterial tests on strains that have not been studied in order to expand the application of M. chinensis in other fields. According to existing results, Staphylococcus aureus can be set as a control group to prove the authenticity of our research data. Due to the concentration setting, the extract did not show any antibacterial effect against fungi, but could exhibit P. acens, MRSA and G. vaginalis. The antibacterial results provide valuable information for developing relevant drugs and cosmetics originated from M. chinensis in the future. This section has been discussed in Results section 2.4 (Line 229-231).

4). The authors in the discussions section could suggest the possible mechanism of action of the extract, commenting on the synergy of the compounds.

Reply: Thank you for your comments. We have dicussed the possible mechanism of action of the main components thymol and carvacrol could destroy the cellular structure (cell membrane) of microorganisms, resulting in the leakage of essential molecules such as protein and K+.  Carvacrol demonstrated relevant clinical antibacterial activity, for both gram-positive and gram negative bacteria, and a synergistic effect when associated with gentamicin (Silva et al., Advances in Sample Preparation, 2023, 7: 10072). We have added the reference in our revised manuscript (Line 259-261, 450-452).
